# Peer review of "Early Enteral Feeding of the Preterm Infant—Delay until Own Mother’s Breastmilk Becomes Available? (Israel, 2012–2017)"

_nutrients, 2022, doi:10.3390/nu14235035_

Round 1
Reviewer 1 Report
The Shlomai et al. article represents a question in those countries where donated milk is not available. Enteral feeding of preterm infants seems to be a challenge between breastfeeding and the trophic requirements of the gastrointestinal system. In this cohort shows that it is necessary to start trophic nutrition (TF), even with preterm formula, as it does not seem to increase the clinical outcome of the neonate. However, some comments I would like to share with the authors.
Introduction:
- Lines 39-40. The citation [1] does not mention increased risk of NEC associated with enteral nutrition. This may be an error.
-Lines 60-63, I suggest that the authors consult the following reviews: PMID: 31185620; PMID: 32092925.
- What was the study hypothesis and objectives?
Material and methods:
- How many neonates were finally recruited? explain in the text. Figure 1 and lines 145-147 should be in the material and methods.
- Define clinical variables: how is hypertension diagnosed, chorioamnionitis, antenatal steroids was it full or partial cycle, was it gender or neonatal sex collected?
- The definition of continuous variables (line 94) is not clear, why was it decided to do? what advantages does it present? line 102-103, were they categorized as "yes" if they presented any of them or was it a summation of 0 = none to 6 = all of them? Clarify in the text.
- Overall, the statistical section is very comprehensive. (line 138) the variables presumed to be associated with the outcome variable, did you follow any significance criteria?
Results:
- Table S1, should be within the main text, these are clinical variables that may explain why certain neonates do not reach defined TF requirements.
- Lines 162-166 should be in methods. Table 1 could be described in the text, since, as a table, it does not add much. However, table 2 is really interesting. It seems that the data goes against in the literature, how do you explain that exclusively breastfed neonates, compared to formula, have more prevalence of BPD, comorbid ratio, more days on mechanical ventilation, increased NICU stay, etc.? could it be because they were smaller and more premature neonates?
- In fact, in this cohort, exclusive breastfeeding could be a risk factor for increased time in IV treatment. I think this is a cohort in which prematurity and comorbidities are shaping the clinical fate of these neonates much more than nutrition. Another important role would be to know the energy breast milk is providing to these neonates, what was the maternal dietary pattern or their body composition. I suggest to consult PMID: 34067287
Discussion:
- Lienas 236-239. I strongly agree with the authors, in fact, I think their data point out that it is important to initiate TF, even by formula in these countries where donor milk is not available. They have already shown that it does not seem to increase comorbidities.
Minor comments:
- Revise grammatical constructions: lines 68-69, line 101, etc.
- Line 67 is not necessary (maybe write "study design"?).
- line 134, write "non-normal" instead of "abnormal".
- line 184, is it amnionitis or chorioaminionitis?
Author Response
Reviewer 1 Introduction: - Lines 39-40. The citation [1] does not mention increased risk of NEC associated with enteral nutrition. This may be an error. We thank the reviewer for revealing this error. The following reference was used BERSETH, Carol Lynn. Feeding strategies and necrotizing enterocolitis. Current opinion in pediatrics, 2005, 17.2: 170-173. -Lines 60-63, I suggest that the authors consult the following reviews: PMID: 31185620; PMID: 32092925. We read with interest the following reviews and added the relevant content to the manuscript Breastmilk is the recommended enteral nutrition for preterm infants as it is associated with improved intestinal development and reduced risk for NEC, with better neurodevelopmental outcomes and with decreased risk for later metabolic and cardiovascular disease. Breastmilk contains multiple bioactive factors such as antioxidants, growth factors, adipokines, and cytokines, as well as unique nutritional factors including fatty acids and fatty acid-derived terminal mediators that provide an energy source as well as regulators of development, immune function, and metabolism. - What was the study hypothesis and objectives? We thank the reviewer for this important comment. We have added accordingly: To the best of our knowledge, the effect of TF content on early neonatal outcomes has not been previously studied. Our primary objective was to assess the effect of TF content on early gastrointestinal morbidities. Secondary outcomes included the effect of TF content on other common early neonatal morbidities e.g. late onset sepsis, duration of parenteral nutrition and death. We hypothesized that TF content would not negatively affect early neonatal outcomes among preterm infants. Materials and methods: - How many neonates were finally recruited? explain in the text. Figure 1 and lines 145-147 should be in the material and methods. The following was removed to “material and methods”: ” During the study period, 608 preterm infants were born at the medical center. Excluded were 97 infants (21 died within the first 72 hours of life, 10 were diagnosed with genetic abnormalities, 66 did not receive TF) (Figure 1). Infants who did not receive TF were born significantly earlier, had lower BW and lower BW z-scores, and were more likely to be born via cesarean section (table 1). The final cohort included 511 infants.” - Define clinical variables: how is hypertension diagnosed, chorioamnionitis, antenatal steroids was it full or partial cycle, was it gender or neonatal sex collected? We have performed the following changes: This data included: maternal and infant characteristics such as maternal hypertension (defined as any hypertension diagnosed during pregnancy), clinical chorioamnionitis (defined as maternal fever accompanied by fetal tachycardia, foul smelling amniotic fluid and abdominal tenderness), antenatal steroids (partial or complete course), multiple pregnancy, gestational age (GA), birth weight (BW), BW z-score, neonatal sex, and Apgar scores - The definition of continuous variables (line 94) is not clear, why was it decided to do? what advantages does it present? We clarified: As the duration of total parenteral nutrition, oxygen requirement, and hospital stay were skewed, they were transformed into categorical variables by defining them as “longer” if levels were above the 66th percentile of the cohort. -line 102-103, were they categorized as "yes" if they presented any of them or was it a summation of 0 = none to 6 = all of them? Clarify in the text. We clarified: Poor composite outcome was defined as “yes” if any of the following occurred: NEC, SIP, bloody stool, sepsis, BPD or death. - Overall, the statistical section is very comprehensive. (line 138) the variables presumed to be associated with the outcome variable, did you follow any significance criteria? We clarified: pReviewer 2 Report
i had already done most of the review. my comments are below. i think they are particularly important because they go to the statistical methodology.
Review of nutrients-2004411-v1
To start, let me state that I am the statistical reviewer and know very little about this subject matter. Please excuse any mistakes I make below.
This is a basically well written paper, but I have questions about the methodology. We basically have a case series in which decisions regarding the exposure of interest were made based on unknown criteria, possibly including the infants’ clinical status. Since studies of this topic are rare and likely to remain so, and unlikely to be randomized, and the outcomes potentially so serious, it is important to do a proper causal analysis. Where I work, this might involve some sort of multinomial or generalized logistic model based on the early characteristics (up to Apgar scores in Table 1), with the predicted probabilities of the 3 arms used to produce (inverse) weights for the observations. After that, the analysis would proceed as the paper does, but using the weights. I gather that social factors were not considered to be relevant. Is this actually true regarding, say, mothers who requested no formula feeding?
It is also possible that the results shown are particular to the setting and time. I therefore request that the title and all tables show the location and years (Israel, 2012-2017).
General comment: It helps the reader to have tables and discussions follow the same order. In this manuscript, the order of Table 2 is EBM vs other, EF vs other, EBM vs EF. The order in Fig. 2 is EF vs other, EBM vs other, EF vs EBM (here, apparently the wording implies the “opposite” of the comparisons in Table 2, but the results look like Table 2). Please fix this.
General question: Did combined EF-EBM occur in a specific sequence (EF first, then EBM)? Is this relevant to the analysis? Would feeding decisions change if trouble happened during the EF phase? If this is an issue, it might be reasonable to do a survival analysis with time-varying variables, rather than treating the data as cross-sectional.
Specific comments:
Line comment
70 delete “, were” at the end.
105 one unit or more, i.e., “Unit’s” or “Units’”?
127-128 “Continuous” not “Continues”
129 I think they mean what is often called a QQ plot. It is possible that nomenclature varies with geography.
140 Arguably, they should designate one primary outcome and do statistical tests for it, while designating all others as secondary and reporting only confidence intervals. The alternative is to take account of multiple testing.
Table 1 Age at discharge—live discharges only? Including deaths probably biases the answer. I’d like to see live discharges (n (%)), ga at live discharge (med (q1, q3) (or do they mean days after birth?), deaths (n (%)), age at death (med (q1-q3)).
Author Response
Reviewer 2 This is a basically well written paper, but I have questions about the methodology. We basically have a case series in which decisions regarding the exposure of interest were made based on unknown criteria, possibly including the infants’ clinical status. Since studies of this topic are rare and likely to remain so, and unlikely to be randomized, and the outcomes potentially so serious, it is important to do a proper causal analysis. Where I work, this might involve some sort of multinomial or generalized logistic model based on the early characteristics (up to Apgar scores in Table 1), with the predicted probabilities of the 3 arms used to produce (inverse) weights for the observations. After that, the analysis would proceed as the paper does, but using the weights. I gather that social factors were not considered to be relevant. Is this actually true regarding, say, mothers who requested no formula feeding? Thank you for your comments This study is an historical cohort study (not a case series study). As you mention, this research question was not studied before. Therefore, we preferred to use the most traditional method, i.e; adjustment using regression modeling to adjust for the potential confounders rather than using inverse probability weighting (IPW), thus creating a pseudo population. Future studies, will probably use more advanced statistical methods. It is also possible that the results shown are particular to the setting and time. I therefore request that the title and all tables show the location and years (Israel, 2012-2017). The manuscript was changed as per reviewer’s recommendation. General comment: It helps the reader to have tables and discussions follow the same order. In this manuscript, the order of Table 2 is EBM vs other, EF vs other, EBM vs EF. The order in Fig. 2 is EF vs other, EBM vs other, EF vs EBM (here, apparently the wording implies the “opposite” of the comparisons in Table 2, but the results look like Table 2). Please fix this. General question: Did combined EF-EBM occur in a specific sequence (EF first, then EBM)? Is this relevant to the analysis? Would feeding decisions change if trouble happened during the EF phase? If this is an issue, it might be reasonable to do a survival analysis with time-varying variables, rather than treating the data as cross-sectional. No there is no sequence. Once the child received at least one feed of each type, it was defined as combined Specific comments: 70 delete “, were” at the end. We changed accordingly 105 one unit or more, i.e., “Unit’s” or “Units’”? During the study period, nutritional guidelines at out institute included 127-128 “Continuous” not “Continues” Continuous 129 I think they mean what is often called a QQ plot. It is possible that nomenclature varies with geography. Q-Q 140 Arguably, they should designate one primary outcome and do statistical tests for it, while designating all others as secondary and reporting only confidence intervals. The alternative is to take account of multiple testing. We have clarified: Our primary objective was to assess the effect of TF content on early gastrointestinal morbidities. Secondary outcomes included the effect of TF content on other common early neonatal morbidities e.g late onset sepsis, duration of parenteral nutrition and death. We hypothesized that TF content would not negatively affect early neonatal outcomes among preterm infants. Table 1 Age at discharge—live discharges only? Including deaths probably biases the answer. I’d like to see live discharges (n (%)), ga at live discharge (med (q1, q3) (or do they mean days after birth?), deaths (n (%)), age at death (med (q1-q3)). The table was corrected accordinglyReviewer 3 Report
Major comments
Authors set out to answer question if early enteral (trophic) feeding of very preterm, low birth weight infants be delayed until the infant's own mother's breastmilk becomes available. They attempted to address the question indirectly by comparing outcomes in babies fed: 1) exclusive breastmilk vs breastmilk + formula, 2) exclusive formula vs formula + breastmilk, 3) exclusive breastmilk vs exclusive formula. They used retrospective data collected on preterm infants born before 32 weeks in a single tertiary center for the analysis. They concluded from their study that "exclusive formula trophic feeding was not associated with an increased risk of any of the studied morbidities".
The introduction was sparse on the benefits of early introduction of enteral feeds, especially of mothers' own breast milk, on the growth and protection of the very preterm infant from morbidities including sepsis and necrotizing enterocolitis. The blanket statement in Lines 38 - 40 "However, enteral nutrition is associated with increased risk for feeding intolerance and necrotizing enterocolitis (NEC) among preterm infants" is not entirely correct and almost misleading. The study aim is stated in the Abstract but not clearly articulated in the Introduction.
Materials & Methods - Was there a difference in age of starting trophic feeds between exclusive breastmilk and exclusive formula fed babies - did use of exclusive breastmilk cause a delay in trophic feeds? The exposures under study were trophic feeds with exclusive breastmilk vs exclusive formula feeds. The benefits of early introduction of trophic feeding, as stated in the introduction, were largely absent from the outcome measures studied.
Transformation of continuous variables into categorical variables - "if levels were above 66th percentile" - appears quite arbitrary. What is the basis for this transformation? How do the authors define "early" parenteral nutrition in their Unit's nutrition guidelines?
Results - The comparison groups with 'clean' exposures were the exclusive breastmilk vs exclusive formula fed babies. It is not possible to draw any conclusion from the other two comparisons as the 'mixed' group has both breastmilk and formula fed babies. The exclusive breastmilk group had statistically significantly lower gestational age and birthweight than exclusive formula fed babies which easily explains the higher proportion of "poor composite outcomes" in the exclusive breastfed babies. Furthermore, the well known benefits of trophic feeds in very low birth weight infants are grouped together in the composite outcome, making it difficult to draw specific conclusions from the results of the study as presented.
The clear benefits of mom's breastmilk over formula feeds and the widespread availability of donor breastmilk (shown to be superior to formula in several studies) seriously diminish the applicability of the findings in many developed countries. Furthermore, the well known dangers (including infections, diarrheal diseases, mortality etc.) of formula feeds in "resource poor" countries also limits the applicability of the findings and recommendations to such countries. The authors indirectly addressed the important issue of initiating trophic feeds in very preterm infants with formula or wait for mom's breast milk. However the use of a retrospective study design, the groups analyzed, the small numbers in the exclusive breastmilk and exclusive formula groups, and the very small numbers in the outcomes of interest detract from the study quality.
Minor comments
Line 69 - delete the word 'were' at the end of the line.
Line 134 - change 'continues' to 'continuous'.
Figure 1 - Study period was between 2012 and 2017, not 2016 and 2017.
Author Response
Reviewer 3 The introduction was sparse on the benefits of early introduction of enteral feeds, especially of mothers' own breast milk, on the growth and protection of the very preterm infant from morbidities including sepsis and necrotizing enterocolitis. We thank the reviewer for his comment. We added the following: Breastmilk is the recommended enteral nutrition for preterm infants,[10] as it is associated with improved intestinal development, reduced risk for NEC, better neurodevelopmental outcomes and with decreased risk for later metabolic and cardiovascular disease.[10] Breastmilk contains multiple bioactive factors such as antioxidants, growth factors, adipokines, and cytokines, as well as unique nutritional factors including fatty acids and fatty acid-derived terminal mediators that provide an energy source as well as regulators of development, immune function, and metabolism. [11-13] The blanket statement in Lines 38 - 40 "However, enteral nutrition is associated with increased risk for feeding intolerance and necrotizing enterocolitis (NEC) among preterm infants" is not entirely correct and almost misleading. Thank you for this important comment. We have changed this as following: Preterm birth is associated with increased risk for feeding intolerance and necrotizing enterocolitis (NEC). The incidence of NEC is inversely related to gestational age and 90% of cases occur after infants have been fed. The study aim is stated in the Abstract but not clearly articulated in the Introduction. Our primary objective was to assess the effect of TF content on early gastrointestinal morbidities. Secondary outcomes included the effect of TF content on other common early neonatal morbidities e.g late onset sepsis, duration of parenteral nutrition and death. We hypothesized that TF content would not negatively affect early neonatal outcomes among preterm infants. Materials & Methods - Was there a difference in age of starting trophic feeds between exclusive breastmilk and exclusive formula fed babies - did use of exclusive breastmilk cause a delay in trophic feeds? The exposures under study were trophic feeds with exclusive breastmilk vs exclusive formula feeds. The benefits of early introduction of trophic feeding, as stated in the introduction, were largely absent from the outcome measures studied. We defined TF as introducing enteral nutrition within the first 72 hours of life. We did not collect data regarding age in hours of 1st TF. We assume that factors such as time of the day, mode of delivery, maternal condition would have much influence on the exact timing. The aim of the study was to assess the effect of TF content, hance we did not progress to compare outcomes of those not exposed to TF. We did assess for sepsis, gastrointestinal complication and time to full enteral feeding (duration of total parenteral nutrition) Transformation of continuous variables into categorical variables - "if levels were above 66th percentile" - appears quite arbitrary. What is the basis for this transformation? We added the reason for this: As the duration of total parenteral nutrition, oxygen requirement, and hospital stay were skewed, they were transformed into categorical variables by defining them as “longer” if levels were above the 66th percentile of the cohort. How do the authors define "early" parenteral nutrition in their Unit's nutrition guidelines? We have specified: “included early parenteral nutrition (i.e; commenced once intravenous access was available) parenteral “ Results The comparison groups with 'clean' exposures were the exclusive breastmilk vs exclusive formula fed babies. It is not possible to draw any conclusion from the other two comparisons as the 'mixed' group has both breastmilk and formula fed babies. The exclusive breastmilk group had statistically significantly lower gestational age and birthweight than exclusive formula fed babies which easily explains the higher proportion of "poor composite outcomes" in the exclusive breastfed babies. Furthermore, the well-known benefits of trophic feeds in very low birth weight infants are grouped together in the composite outcome, making it difficult to draw specific conclusions from the results of the study as presented. This is correct, we discussed this: “The exclusive breastmilk group differed significantly from the other groups in that the infants were born earlier, were lighter, and had lower Apgar scores. This group also received a significantly lower amount of TF during the first 72 hours of life. As expected, this group was more likely to be diagnosed with BPD, poor composite outcome, longer hospital stays and longer duration of intravenous fluid treatment. Yet the incidence of gastrointestinal complications or sepsis did not differ between the groups. Multivariate analysis correcting for gender, GA, BW-z score, amnionitis, Ibuprofen treatment and antenatal steroids exposure did not demonstrate any differences in early neonatal morbidities, with the exception of longer duration of parenteral nutrition among the exclusive breastmilk group. Specifically, the exclusive formula group was not shown to be at increased risk for any of the studied morbidities. Therefore, we concluded that in the present cohort, TF content given during the first 72 hours of life was not shown to affect the risk for early neonatal morbidities or mortality.” The clear benefits of mom's breastmilk over formula feeds and the widespread availability of donor breastmilk (shown to be superior to formula in several studies) seriously diminish the applicability of the findings in many developed countries. Furthermore, the well-known dangers (including infections, diarrheal diseases, mortality etc.) of formula feeds in "resource poor" countries also limits the applicability of the findings and recommendations to such countries. The authors indirectly addressed the important issue of initiating trophic feeds in very preterm infants with formula or wait for mom's breast milk. However, the use of a retrospective study design, the groups analyzed, the small numbers in the exclusive breastmilk and exclusive formula groups, and the very small numbers in the outcomes of interest detract from the study quality. We thank you for this comment. This study was based in data of the 3rd largest hospital in Israel with a relatively recent cohort. The conflict of feeding with formula or waiting for BM is relevant for countries that do not have donor milk but also for population that refuse using donor milk. We believe that the presented data may help in decision making in such cases. Minor comments Line 69 - delete the word 'were' at the end of the line. Was removed Line 134 - change 'continues' to 'continuous'. Was changed Figure 1 - Study period was between 2012 and 2017, not 2016 and 2017. Revised according to the correct dates.Round 2
Reviewer 1 Report
Thank you for this revised version of the manuscript. However, I have not been able to identify the changes in the main text because they are not underlined or color-coded. On the other hand, the authors' report does not respond to the comments in the results and discussion sections.
Author Response
We apologize for sending only part of the replay to your review. Please find attached our response to the whole review Introduction. The changes in the main text with color-coded were submitted earlier and are attached below, too (author-coverletter). An extensive editing of the English language was done as well
- Overall, the statistical section is very comprehensive. (line 138) the variables presumed to be associated with the outcome variable, did you follow any significance criteria?
We clarified; now lines 161-162:
, and p <0.05 was considered to be statistically significant throughout the study
Results:
- Table S1, should be within the main text, these are clinical variables that may explain why certain neonates do not reach defined TF requirements.
Table S1 was moved to the main text and is now called Table 1. Now line 210
- Lines 162-166 should be in methods. Table 1 could be described in the text, since, as a table, it does not add much. However, table 2 is really interesting. It seems that the data goes against in the literature, how do you explain that exclusively breastfed neonates, compared to formula, have more prevalence of BPD, comorbid ratio, more days on mechanical ventilation, increased NICU stay, etc.? could it be because they were smaller and more premature neonates?
Thank you, we explained our hypothesis in the discussion part, lines 249-254:
The infants in the exclusive breastmilk group differed significantly from those in other groups in that they were born earlier, had lower birth weights, and had lower Apgar scores. This group also received a significantly smaller amount of TF during the first 72 hours of life. As expected, this group was more likely to be diagnosed with BPD, poor composite outcome, longer hospital stays and longer duration of intravenous fluid treatment.
- In fact, in this cohort, exclusive breastfeeding could be a risk factor for increased time in IV treatment. I think this is a cohort in which prematurity and comorbidities are shaping the clinical fate of these neonates much more than nutrition. Another
We have discussed this problem in the discussion part and conducted a multivariate analysis correcting for gender, GA, BW-z score, chorioamnionitis: lines 249-261
The infants in the exclusive breastmilk group differed significantly from those in other groups in that they were born earlier, had lower birth weights, and had lower Apgar scores. This group also received a significantly smaller amount of TF during the first 72 hours of life. As expected, this group was more likely to be diagnosed with BPD, poor composite outcome, longer hospital stays and longer duration of intravenous fluid treatment. Nevertheless, the incidence of gastrointestinal complications or sepsis did not differ between the groups. Multivariate analysis correcting for gender, GA, BW-z score, amnionitis, ibuprofen treatment and exposure to antenatal steroids did not reveal any differences in early neonatal morbidities, with the exception of longer duration of parenteral nutrition among the exclusive breastmilk group.
- important role would be to know the energy breast milk is providing to these neonates, what was the maternal dietary pattern or their body composition. I suggest to consult PMID: 34067287
Unfortunately, this data was not available to us and cannot be retrospectively collected, however this was added to the limitation part of our manuscript; lines 312-314
Moreover, the study did not assess maternal dietary patterns or maternal body composition, which have been shown to influence breastmilk compounds and newborns’ health. (24)
- Lienas 236-239. I strongly agree with the authors, in fact, I think their data point out that it is important to initiate TF, even by formula in these countries where donor milk is not available. They have already shown that it does not seem to increase comorbidities.
Thank you
Minor comments:
- Revise grammatical constructions: lines 68-69, line 101, etc.
Changes were made:
Eligible for the study were infants born < 32 weeks of gestation between the years 2012- 2017 in a single tertiary medical center.
For the purpose of statistical analysis, unfavorable outcomes (BPD, longer oxygen requirement etc.) and death (as we couldn’t exclude these outcomes in infants who died prior to discharge), were calculated as a composite of the prespecified outcome.
- Line 67 is not necessary (maybe write "study design"?).
We changed now line 78: "study design"
- line 134, write "non-normal" instead of "abnormal".
We corrected to "non-normal" , line 155
- line 184, is it amnionitis or chorioaminionitis?
We corrected, now line 205 "chorioaminionitis"
